# VLEARN: OFF-POLICY LEARNING WITH EFFICIENT STATE-VALUE-FUNCTION ESTIMATION

## ABSTRACT

Existing off-policy reinforcement learning algorithms typically necessitate an explicit state-action-value function representation, which becomes problematic in high-dimensional action spaces. These algorithms often encounter challenges where they struggle with the curse of dimensionality, as maintaining a state-action-value function in such spaces becomes data-inefficient. In this work, we propose a novel off-policy trust region optimization approach, called Vlearn, that eliminates the requirement for an explicit state-action-value function. Instead, we demonstrate how to efficiently leverage just a state-value function as the critic, thus overcoming several limitations of existing methods. By doing so, Vlearn addresses the computational challenges posed by high-dimensional action spaces. Furthermore, Vlearn introduces an efficient approach to address the challenges associated with pure state-value function learning in the off-policy setting. This approach not only simplifies the implementation of off-policy policy gradient algorithms but also leads to consistent and robust performance across various benchmark tasks. Specifically, by removing the need for a state-action-value function Vlearn simplifies the learning process and allows for more efficient exploration and exploitation in complex environments.

## 1 INTRODUCTION

*Reinforcement learning* (RL) has emerged as a powerful paradigm for training intelligent agents through interaction with their environment (Mnih et al., 2013; Silver et al., 2016). Within RL, there exist two primary approaches for model-free learning: on-policy and off-policy methods. On-policy methods rely on newly generated (quasi-)online samples in each iteration (Schulman et al., 2017; Otto et al., 2021), whereas off-policy methods leverage a replay buffer populated by a behavior policy (Abdolmaleki et al., 2018; Haarnoja et al., 2018; Fujimoto et al., 2018). Although on-policy methods can compensate for stale/off-policy data to some extent via importance sampling Espeholt et al. (2018), they are still not able to fully exploit it.

To harness the full potential of off-policy data, off-policy methods traditionally focus on learning state-action-value functions (Q-functions) as critics (Degris et al., 2012). The Q-function's dependency on the state as well as the action enables it to update only those actions that have been observed in the transitions generated by the behavior policy. However, the complexity associated with learning Q-functions, especially in high-dimensional action spaces, is often undesirable, and alternatives similar to the on-policy setting based on state-value functions (V-functions), would be preferable.

In this work, we introduce Vlearn, a novel approach to off-policy policy gradient learning that exclusively leverages V-functions. While existing methods, such as V-trace (Espeholt et al., 2018), have tried to increase the amount of stale data on-policy methods can exploit, we found them to struggle in a full off-policy setting. In particular, they only aim to reweigh the Bellman targets. Yet, previous work (Mahmood et al., 2014) has already shown that importance sampling for the full Bellman error is preferable to such reweighing of the Bellman targets. Our method optimizes an upper bound of the original Bellman error, which can be derived using Jensen's inequality. This bound effectively shifts the importance weights from the Bellman targets to the optimization objective itself, which simplifies V-function updates and increases the stability of learning a V-function from off-policy data, a hallmark of previous approaches. In addition to this advancement, we further enhance the stability of policy learning by combining it with the trust region update introduced by Otto et al.

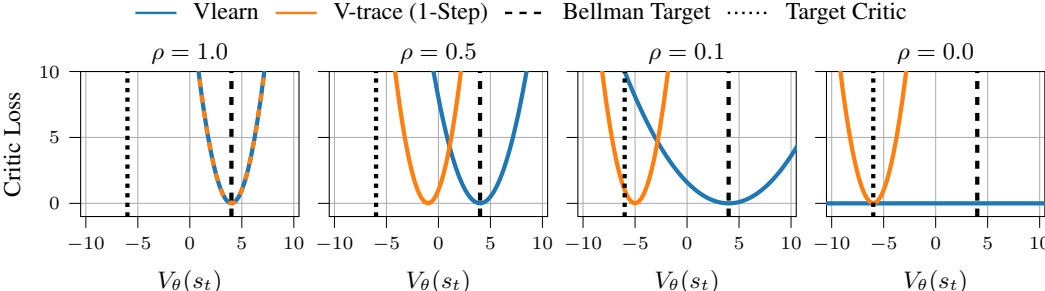

Figure 1: To provide intuition on the differences between Vlearn and V-trace we consider the following example. Assume that for a state $s_t$, the Bellman target is $r(s, a) + \gamma V_{\bar{\theta}}(s_{t+1}) = 4$, the target critic predicts $V_{\bar{\theta}}(s_t) = -6$ and we plot the loss for potential values of $V_{\theta}(s)$ for Vlearn and V-trace. For on-policy samples ($\rho = 1.0$) both losses are the same. However, for samples that are more and more off-policy ($\rho \to 0$) we see how V-trace increasingly relies on the target critic, shifting the optimal value towards it. Vlearn on the other hand simply reduces the scale of the loss and thus the importance of the sample. This makes Vlearn more robust to errors in the target critic.

(2021), forming an efficient off-policy trust region method. In our experiments, we demonstrate the benefits of this approach, especially in environments with high-dimensional action spaces, such as the notoriously difficult dog locomotion tasks of *DeepMind control* (DMC), which currently cannot be solved by most standard off-policy actor-critic methods.

## 2    RELATED WORK

To enhance sample efficiency in RL, off-policy algorithms aim to leverage historical data stored in a replay buffer (Lin, 1992). While deep Q-Learning-based methods (Mnih et al., 2013; Van Hasselt et al., 2016; Wang et al., 2016; Hessel et al., 2018) excel in efficiently learning from discrete action spaces, they are often not directly applicable to continuous problems. For such problems, off-policy actor-critic methods optimize a policy network that generates continuous action values based on the state-action-value function estimator (Degris et al., 2012; Zhang et al., 2019). This approach accommodates both deterministic (Lillicrap et al., 2015; Fujimoto et al., 2018) or probabilistic policy distributions (Haarnoja et al., 2018; Abdolmaleki et al., 2018). Nevertheless, similar to deep Q-learning, training a state-action-value function estimator can be challenging and often presents similar difficulties as in the discrete setting (Fujimoto et al., 2018). Here, some solution strategies directly transfer from the discrete to the continuous settings (Fujimoto et al., 2018). We can alleviate some of these issues by resorting to common techniques, such as ensembles, to further improve the performance (Chen et al., 2021)

In the on-policy setting, trust region methods (Schulman et al., 2017; 2015; Otto et al., 2021) have proven effective in stabilizing the policy gradient. However, these methods have not received the same level of attention in the off-policy setting. Past efforts have focused on training the value function with off-policy data (Gu et al., 2016) or extended trust region policy optimization (Schulman et al., 2015) to the off-policy setting (Nachum et al., 2018; Meng et al., 2022). While Wang et al. (2017), for instance, employs a more standard approach by combining off-policy trust regions with additional advancements, such as Retrace (Munos et al., 2016) and truncated importance sampling, it is still not able to compete with modern actor-critic approaches. Peng et al. (2019), on the other hand, provide a different perspective and split policy and value-function learning into separate supervised learning steps. Generally, standard off-policy trust region methods often fall behind modern actor-critic approaches (Haarnoja et al., 2018; Fujimoto et al., 2018) and cannot achieve competitive performance. However, *maximum aposteriori policy optimisation* (MPO) (Abdolmaleki et al., 2018) could be seen as a trust region method in a non-classical sense. Its EM-based formulation offers more flexibility, with trust regions being akin to optimizing a parametric E-step without an explicit M-step.

While in the on-policy setting trust regions methods resort to estimating the less complex state-value function, in the off-policy settings, most methods still rely on estimating state-action-value func-

tions to efficiently learn from replay buffer data. Methods exclusively utilizing state-value functions must employ importance sampling to address the distributional discrepancy between target and behavior policies. This is often done by reweighting and truncating the importance of the Bellman targets of n-step returns (Espeholt et al., 2018; Luo et al., 2020) and has even shown to be beneficial when learning a state-action-value-function (Munos et al., 2016). However, this form of off-policy correction is computationally expensive as it relies on storing and processing full/partial trajectories. Moreover, it is widely acknowledged that the truncation technique used in importance weight calculations suffers from a substantial bias. Consequently, the implementation of trajectory-based target estimators would exacerbate the already pronounced issue of bias propagation. On the other hand, previous work demonstrated that using importance sampling on the targets is not optimal and importance weighting the full Bellman error, similar as for the policy gradient, is more desirable (Mahmood et al., 2014; Dann et al., 2014).

## 3 EFFICIENT STATE-VALUE FUNCTION LEARNING FROM OFF-POLICY DATA

We seek to determine an optimal policy $\pi(a|s)$ within the framework of a *Markov decision process* (MDP), which is defined by a tuple $(\mathcal{S}, \mathcal{A}, \mathcal{T}, r, \rho_0, \gamma)$. Here, both the state space $\mathcal{S}$ and action space $\mathcal{A}$ are continuous. The transition density function $\mathcal{T} : \mathcal{S} \times \mathcal{A} \times \mathcal{S} \to \mathbb{R}^+$ is defined as a function mapping from the current state $s_t \in \mathcal{S}$ and action $a_t \in \mathcal{A}$ to the probability density of transitioning to the subsequent state $s_{t+1} \in \mathcal{S}$. The initial state distribution's density is denoted as $\rho_0 : \mathcal{S} \to \mathbb{R}^+$. The reward attained from interactions with the environment is determined by the function $r : \mathcal{S} \times \mathcal{A} \to \mathbb{R}$, and the parameter $\gamma \in [0, 1)$ represents the discount factor applied to future rewards. Our primary objective is to maximize the expected cumulative discounted reward

$$G_t = \mathbb{E}_{\mathcal{T}, \rho_0, \pi} \left[ \sum_{k=t}^{\infty} \gamma^{k-t} r(s_k, a_k) \right].$$

Most popular off-policy actor-critic methods (Haarnoja et al., 2018; Fujimoto et al., 2018; Abdolmaleki et al., 2018) now aim to find a policy that maximizes the cumulative discounted reward by making use of a learnable state-action value estimate $Q_\theta^\pi(s, a) = \mathbb{E}_\pi[G_t|s_t = s, a_t = a]$. To train this estimator, they rely on a dataset $\mathcal{D} = \{(s_t, a_t, r_t, s_{t+1})_{t=1...N}\}$ and a behavioral policy $\pi_b(\cdot|s)$ responsible for generating this dataset. Typically, $\mathcal{D}$ takes the form of a replay buffer (Lin, 1992), with the corresponding behavior policy $\pi_b$ being a mixture of the historical policies used to populate the buffer. Training the state-action value function then usually involves temporal difference learning (Sutton, 1988; Watkins & Dayan, 1992), with updates grounded in the Bellman equation (Bellman, 1957). The objective commonly optimized is

$$J(\theta) = \mathbb{E}_{(s_t, a_t) \sim \mathcal{D}} \left[ \left( Q_\theta^\pi(s_t, a_t) - \left( r(s_t, a_t) + \gamma \mathbb{E}_{s_{t+1} \sim \mathcal{D}, a_{t+1} \sim \pi(\cdot|s_{t+1})} Q_{\bar{\theta}}(s_{t+1}, a_{t+1}) \right) \right)^2 \right], \quad (1)$$

where $Q_{\bar{\theta}}(s, a)$ is a frozen target network. Furthermore, to mitigate overestimation bias, most methods employ two state-action value functions (Fujimoto et al., 2018). To maintain the stability of this approach, the target network's weights are updated at each time step as $\bar{\theta} \leftarrow \tau\theta + (1 - \tau)\bar{\theta}$.

Instead of using the state-action function $Q_\theta^\pi(s_t, a_t)$, an alternative approach is to work solely with the state-value function $V_\theta^\pi(s_t)$. This estimator can be trained by minimizing the following loss function using importance sampling

$$J(\theta) = \mathbb{E}_{s_t \sim \mathcal{D}} \left[ \left( V_\theta^\pi(s_t) - \mathbb{E}_{(a_t, s_{t+1}) \sim \mathcal{D}} \left[ \frac{\pi(a_t|s_t)}{\pi_b(a_t|s_t)} \left( r(s_t, a_t) + \gamma V_{\bar{\theta}}(s_{t+1}) \right) \right] \right)^2 \right]. \quad (2)$$

Directly optimizing this can be difficult as approximating the inner expectation with just one Monte Carlo sample yields a high variance. Achieving a more reliable estimate necessitates multiple samples, implying either multiple action executions per state or occurrences of the same state in the replay buffer, an unrealistic assumption for most RL problems. An essential distinction from the Q-function based objective in Equation 1 lies in the introduction of the importance weight $\pi(a_{t,j}|s_t)/\pi_b(a_{t,j}|s_t)$, which accounts for the difference between the behavior distribution $\pi_b(\cdot|s)$ and the current policy $\pi(\cdot|s)$. Unlike the Q-function, which updates its estimate solely for the chosen action, the V-function lacks a dependency on the action and hence implicitly updates its estimate for all actions. As a result, we need to consider the difference between the current policy and the policy that collected the data.

The objective in Equation 2 bears similarity to the 1-step V-trace estimate (Espeholt et al., 2018). It can be seen as a naive version of V-trace because a large difference between the target and behavior policies may result in importance weights approaching either zero or infinity, consequently impacting the Bellman optimization target for the state-value function $V_\theta^\pi(s_t)$. While truncated importance weights can prevent excessively large values for the Bellman target, the importance ratios and target may still approach values close to zero. V-trace reformulates this objective and effectively interpolates between the Bellman target and the current (target) value function in the one step case $n = 1$

$$J(\theta) = \mathbb{E}_{s_t \sim \mathcal{D}} \left[ \left( V_\theta^\pi(s_t) - \left( (1 - \rho_t) V_{\bar{\theta}}(s_t) + \rho_t \left( r_t + \gamma V_{\bar{\theta}}(s_{t+1}) \right) \right) \right)^2 \right], \qquad (3)$$

where $\rho_t = \min(\pi(a_t|s_t)/\pi_b(a_t|s_t), \epsilon_\rho)$ are the truncated importance weights with a user specified upper bound (typically $\epsilon_\rho = 1$). This objective can then also be extended for off-policy corrections of n-step returns (Espeholt et al., 2018). Yet, this formulation has two main drawbacks. Since executing multiple actions for the same state is impractical, or even impossible, V-trace approximates the inner expectation with just one action sample, resulting in potentially undesirable high variance estimates. Moreover, while the V-trace reformulation avoids that the value estimate is optimized towards zero for small importance ratios, the interpolation now optimizes it increasingly towards the current (target) value function. This interpolation also leads to a shift of the optimum, which means for samples with small importance ratios the value function is barely making any learning progress and maintains its current estimate. Important to note, only the optimum changes; the scale of the loss function and, hence, the gradient scale remains the same for all importance ratios. In the original work, this issue is less pronounced, as V-trace provides off-policy corrections for asynchronous distributed workers. In this setup, the policy distributions are expected to stay relatively close, mitigating the impact of any potential divergence. However, in a complete off-policy setting, the samples in the replay buffer and the current policy deviate significantly faster and stronger from each other. The size of the replay buffer, learning speed, or entropy of different policies can all influence the importance ratio. This behavior for various importance ratios is illustrated in Figure 1.

As the original V-trace algorithm is primarily utilized for off-policy corrections for a distributed on-policy learning method, it normally does not incorporate a target network. In contrast, it is common practice for off-policy methods (Haarnoja et al., 2018; Fujimoto et al., 2018; Abdolmaleki et al., 2018) to incorporate a target network. This practice is also followed by V-trace's predecessor, Retrace (Munos et al., 2016), and, as a result, we have included it here as well.

A significant challenge with V-trace lies in its use of importance sampling exclusively for the target computations, which can additionally become increasingly costly to compute with larger n-step returns. Moreover, it has already been shown for the linear case that applying the importance weights to just the Bellman targets is inferior to applying them to the full loss (Mahmood et al., 2014; Dann et al., 2014). However, contradicting to these previous findings, the former approach remains the main method in the deep learning setting (Munos et al., 2016; Espeholt et al., 2018). To address the above mentioned issues, we propose a more effective approach to training the value function by following Mahmood et al. (2014); Dann et al. (2014) and shifting the importance ratio to the full loss function by extending their results to the general function approximation case.

**Theorem 1** *Consider the following loss, minimized with respect to the V-function's parameters $\theta$*

$$L(\theta) = \mathbb{E}_{s_t \sim \mathcal{D}} \left[ \mathbb{E}_{(a_t, s_{t+1}) \sim \mathcal{D}} \left[ \frac{\pi(a_t|s_t)}{\pi_b(a_t|s_t)} \left( V_\theta^\pi(s_t) - \left( r_t + \gamma V_{\bar{\theta}}(s_{t+1}) \right) \right)^2 \right] \right].$$

$$= \mathbb{E}_{(s_t, a_t, s_{t+1}) \sim \mathcal{D}} \left[ \frac{\pi(a_t|s_t)}{\pi_b(a_t|s_t)} \left( V_\theta^\pi(s_t) - \left( r_t + \gamma V_{\bar{\theta}}(s_{t+1}) \right) \right)^2 \right]. \qquad (4)$$

*This objective serves as an upper bound to the importance-weighted Bellman Loss (Equation 2). Furthermore, this upper bound is consistent; that is, a value function minimizing Equation 4 also minimizes Equation 2.*

The first statement's proof relies on Jensen's Inequality, while the proof of the second statement is an extension of the work from Neumann & Peters (2008). The complete proofs are provided in Appendix A.1 and Appendix A.2, respectively.

This upper bound of the loss function closely resembles the standard state-action-value loss functions used in other off-policy methods but is based on the action-independent state-value function.

---

**Algorithm 1**

---

1: Initialize policy $\phi$, critics $\theta_1, \theta_2$, target critics $\bar{\theta}_1, \bar{\theta}_2$
2: Initialize replay buffer $\mathcal{D}$, truncation level $\epsilon_\rho$, trust region bounds $\epsilon_\mu, \epsilon_\Sigma$
3: Initialize $\pi_{\text{old}} \leftarrow \pi_\phi$
4: **for** $i = 0, 1, \ldots, N$ **do** $\qquad\qquad\qquad\qquad\qquad\qquad\qquad\qquad\qquad\qquad$ ▷ epoch
5: $\quad$ **for** $j = 0, 1, \ldots, M$ **do**
6: $\qquad$ Collect sample $(s, a, r, s', \pi_\phi(a|s))$ and add to $\mathcal{D}$
7: $\qquad$ Sample batch $\mathcal{B} = \{(s, a, r, s', \pi_b(a|s))\}_{t=1\ldots K}$ from $\mathcal{D}$
8: $\qquad$ Get current policy $\pi_\phi(a|s)$ for all $s$ in $\mathcal{B}$
9: $\qquad$ Compute projected policy $\tilde{\pi}_\phi = \text{TRPL}(\pi_\phi, \pi_{\text{old}}, \epsilon_\mu, \epsilon_\Sigma)$ for all $s$ in $\mathcal{B}$
10: $\qquad$ Update all critic networks with gradient

$$\nabla_{\theta_i} \frac{1}{K} \sum_{\mathcal{B}} \min\left(\frac{\tilde{\pi}_\phi(a|s)}{\pi_b(a|s)}, \epsilon_\rho\right) \left(V_{\theta_i}^\pi(s) - \left(r + \gamma V_{\bar{\theta}_i}(s')\right)\right)^2 \text{ for } i \in \{1, 2\}$$

11: $\qquad$ **if** update policy **then**
12: $\qquad\quad$ Compute advantage estimates $\hat{A} = r + \gamma \min_{i=1,2} V_{\theta_i}^\pi(s') - \min_{i=1,2} V_{\theta_i}^\pi(s)$
13: $\qquad\quad$ Update policy with trust region loss

$$\nabla_\phi \left[\frac{1}{K} \sum_{\mathcal{B}} \left[\min\left(\frac{\tilde{\pi}_\phi(a|s)}{\pi_b(a|s)}, \epsilon_\rho\right) \hat{A}\right] - \alpha \mathrm{d}(\pi_\phi, \tilde{\pi})\right]$$

14: $\qquad$ $\bar{\theta}_i \leftarrow \tau\theta_i + (1-\tau)\bar{\theta}_i$ for $i \in \{1, 2\}$
15: $\qquad$ $\pi_{\text{old}} \leftarrow \pi_\phi$

---

It introduces importance weights to account for the mismatch between behavior and policy distribution. Importantly, evaluating Equation 4 becomes straightforward using the provided replay buffer $\mathcal{D}$. When we use samples from the joint state-action distribution, dealing with only one action per state becomes more manageable. This helps to reduce variance compared to the original and V-trace objective. Unlike V-trace, each sample optimizes towards the Bellman target but has an impact on the total loss per step depending on its importance weight, as shown in Figure 1. The smaller importance weights mainly affect how the gradient is scaled, without causing a shift in optimizing to a different optimum, such as the current (target) value function. This approach makes learning the state-value function in an off-policy setting more stable and efficient.

An essential consideration is defining the behavior policy $\pi_b$ in an off-policy setting, where $\pi_b$ is a mixture of all past policies that contributed to the replay buffer. Computationally, storing and evaluating this mixture policy would be expensive. However, since we rely on importance sampling, which only requires access to the (log-)probabilities of each action, we can easily extend the replay buffer with this single entry at minimal additional cost. In addition, this (log-)probability can then be used directly for importance sampling in the policy update step.

## 3.1 Off-Policy Policy Gradient using the State-Value-Function

To find the optimal policy, conventional policy gradient techniques frequently use the gradient of the likelihood ratio and an importance sampling estimator to optimize an estimate of the Q-function. In particular, a more effective approach then involves optimizing the advantage function, denoted as $A^\pi(s, a) = Q^\pi(s, a) - V^\pi(s)$. Adding the V-function as a baseline yields an unbiased gradient estimator with reduced variance. The optimization can then be formulated as follows

$$\max_\phi \hat{J}(\pi_\phi, \pi_b) = \max_\phi \mathbb{E}_{(s,a)\sim\mathcal{D}}\left[\frac{\pi_\phi(a|s)}{\pi_b(a|s)} A^\pi(s, a)\right]. \tag{5}$$

In the on-policy case, the advantage values are typically estimated via Monte Carlo approaches, such as general advantage estimation (Schulman et al., 2016). Yet, in our off-policy setting, we are neither able to compute a good Monte Carlo estimate, nor can we rely on the Q-function estimate Degris et al. (2012). Further, the direct use of the V-function to improve the policy is not feasible. However, in conjunction with the replay buffer, a strategy akin to the standard policy gradient becomes viable. This approach allows us to employ an off-policy estimate of the V-function, offering a substantial

boost in sample efficiency. The advantage estimate is computed using the one-step return of our off-policy evaluated value function $A^\pi(s_t, a_t) = r_t + \gamma V_\theta^\pi(s_{t+1}) - V_\theta^\pi(s_t)$. The above objective can then be optimized by any policy gradient algorithm, e. g., by *proximal policy optimization* (PPO) (Schulman et al., 2017) or via the *trust region projection layer* (TRPL) (Otto et al., 2021).

In this work, we use TRPL (Otto et al., 2021) as it has been shown to stabilize learning even for complex and high-dimensional action spaces (Otto et al., 2022; 2023). Unlike PPO, it provides a mathematically sound and scalable approach to enforce trust regions exactly per state. Moreover, TRPL allows us to use the constraint policy also during the value function update, which now requires importance sampling. PPO has only the clipped objective for the policy update, and using a clipped value function has been shown to potentially degrade performance (Engstrom et al., 2020).

TRPL efficiently enforces a trust region for each input state of the policy using differentiable convex optimization layers (Agrawal et al., 2019), providing more stability and control during training and at the same time reduce the dependency on implementation choices (Engstrom et al., 2020). Intuitively, the layer ensures the predicted Gaussian distribution from the policy network is always satisfying the trust region constraint. This way, the objective from Equation 5 can directly be optimized as the trust region always holds. The layer receives the network's initial prediction for the mean $\boldsymbol{\mu}$ and covariance $\boldsymbol{\Sigma}$ of a Gaussian distribution, and projects them into the trust region when either exceeds their respective bounds. This is done individually for each state provided to the network. The resulting Gaussian policy distribution, characterized by the projected parameters $\tilde{\boldsymbol{\mu}}$ and $\tilde{\boldsymbol{\Sigma}}$, is then used for subsequent computations, e. g. sampling and loss computation. Formally, as part of the forward pass the layer solves the following two optimization problems for each state $\boldsymbol{s}$

$$\arg\min_{\tilde{\boldsymbol{\mu}}_s} d_{\text{mean}}\left(\tilde{\boldsymbol{\mu}}_s, \boldsymbol{\mu}(s)\right), \quad \text{s.t.} \quad d_{\text{mean}}\left(\tilde{\boldsymbol{\mu}}_s, \boldsymbol{\mu}_{\text{old}}(s)\right) \leq \epsilon_{\boldsymbol{\mu}}, \quad \text{and}$$

$$\arg\min_{\tilde{\boldsymbol{\Sigma}}_s} d_{\text{cov}}\left(\tilde{\boldsymbol{\Sigma}}_s, \boldsymbol{\Sigma}(\boldsymbol{s})\right), \quad \text{s.t.} \quad d_{\text{cov}}\left(\tilde{\boldsymbol{\Sigma}}_s, \boldsymbol{\Sigma}_{\text{old}}(\boldsymbol{s})\right) \leq \epsilon_{\Sigma},$$

where $\tilde{\boldsymbol{\mu}}_s$ and $\tilde{\boldsymbol{\Sigma}}_s$ are the optimization variables for input state $\boldsymbol{s}$. The trust region bounds $\epsilon_\mu$, $\epsilon_\Sigma$ are for the mean and covariance of the Gaussian distribution, respectively. For the dissimilarities between means $d_{\text{mean}}$ and covariances $d_{\text{cov}}$, we use the decomposed KL-divergence. How to receive the gradients for the backward pass via implicit differentiation is described in Otto et al. (2021).

## 3.2 BEHAVIOR POLICY VS OLD POLICY

For Vlearn we keep track of three different policies, the current policy $\pi_\phi$ that is optimized, the old policy $\pi_{\text{old}}$ that is used as reference for the trust region, and the behavior policy $\pi_b$ that is required for the off-policy correction using importance sampling. Similar to most on-policy trust region methods, we maintain $\pi_{\text{old}}$ as a copy of the main policy network from the previous iteration. Choosing the behavioral policy for the trust region would be detrimental to performance because it would slow down the policy network update or, in the worst case, force the policy back to a much older and worse policy distribution. Conversely, we cannot use the copied old policy as a behavior policy because the actual behavior policy $\pi_b$ can be arbitrarily far away, especially for older samples. Generally, the behavior policy is a mixture of all past policies that have been used to populate the replay buffer. However, storing and/or evaluating this would be expensive, hence we make the assumptions that each sample belongs to only one mixture component, in particular, the policy that originally created the action. Since we rely on importance sampling for the off-policy correction, we can simply store the (log-)probability for each action as part of the replay buffer to represent $\pi_b$ during training.

To stabilize our method further, we make use of improvements of other actor-critic and policy gradient methods, including twin critics and delayed policy updates (Fujimoto et al., 2018), $\tanh$ squashing (Haarnoja et al., 2018), as well as advantage normalization (Schulman et al., 2017; Otto et al., 2021). While the overestimation bias is not directly a problem when using just state-value functions, we found them to be beneficial in practice. We assume the twin network can be seen as a small ensemble that provides a form of regularization. Given there is no significant drawback and the widespread adoption of the twin network approach in other baseline methods, we have chosen to maintain its use in our case as well. Additionally, similar to prior works (Munos et al., 2016; Espeholt et al., 2018) we aim to reduce the variance by replacing the standard importance sampling ratio with truncated importance sampling $\min(\pi(a_t|s_t)/\pi_b(a_t|s_t), \epsilon_\rho)$. An overview of our full approach is shown in Algorithm 1.

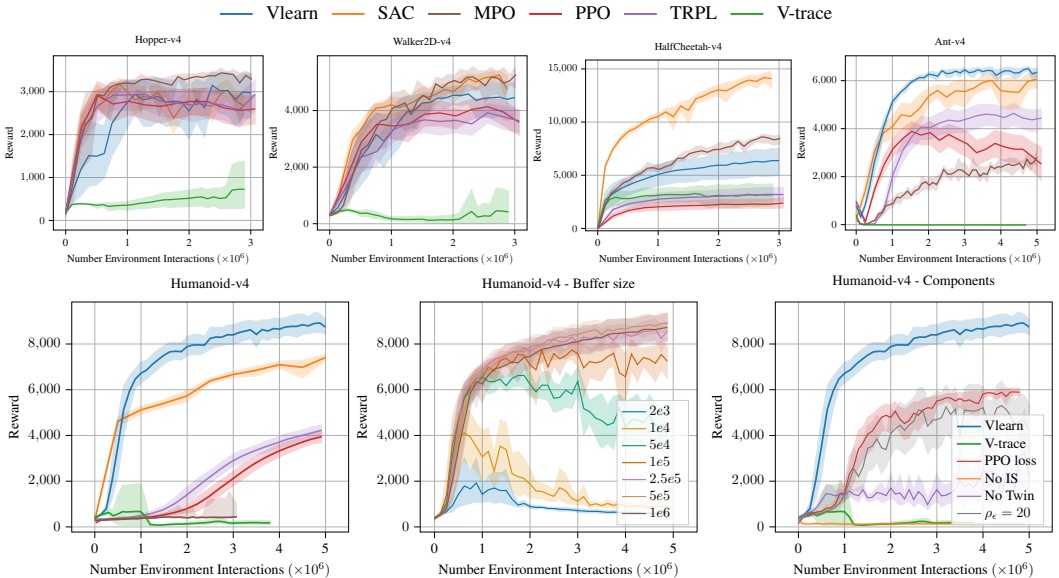

Figure 2: Shown are the mean over 10 seeds and 95% bootstrapped confidence intervals for the Gymnasium tasks. While SAC performs well on the lower dimensional tasks, Vlearn achieves a better asymptotic performance for the higher dimensional problems - Ant-v4 and Humanoid-v4. In comparison to V-trace, our method learns significantly more stable while also achieving an overall better performance. The figure in the bottom center shows an ablation study that compares the effect the replay buffer size has on the policy's performance. For smaller replay buffers learning becomes unstable or does not convergence, while larger sizes tend to lead to a similar final performances. The figure in bottom right shows an ablation study for various variations of our proposed method.

## 4 EXPERIMENTS

For our experiments, we evaluate Vlearn on a variety of different continuous control tasks from Gymnasium (Towers et al., 2023) and DMC (Tunyasuvunakool et al., 2020). As baselines, we trained the standard off-policy methods SAC (Haarnoja et al., 2018) and MPO (Abdolmaleki et al., 2018) as well as the original on-policy version of TRPL (Otto et al., 2021) and PPO (Schulman et al., 2017). We additionally, compare our method to 1-step V-trace (Espeholt et al., 2018) by replacing our lower bound objective from Equation 4 with the objective in Equation 3, which we refer to as *V-trace*. This comparison aims to emphasize the difference in the placement of importance ratios. To ensure a fair assessment, we aim to eliminating all external factors that could influence the results and, hence, do not use n-step returns. While n-step V-trace can be computed recursively, our approach could leverages the product of the n-step importance weights, preserving consistency with V-trace. All other components of the method are kept the same as for Vlearn.

In terms of visualization, we follow the approach outlined in Agarwal et al. (2021). All methods are evaluated for 10 different seeds each, and their performance is aggregated using mean values and 95% bootstrapped confidence intervals for individual tasks. We maintain uniformity in the architecture of policy and critic networks for all methods, incorporating layer normalization (Ba et al., 2016) before the first hidden layer. Hyperparameters are held constant, and only appropriately adjusted for the higher-dimensional dog tasks. Detailed hyperparameter information for all methods can be found in Appendix B.

### 4.1 GYMNASIUM

The results for the Gymnasium tasks are shown in Figure 2. Vlearn shows good performance on lower dimensional tasks, achieving asymptotic performance comparable to SAC, with the exception of HalfCheetah-v4. While Vlearn generally outperforms on-policy methods on the HalfCheetah-v4,

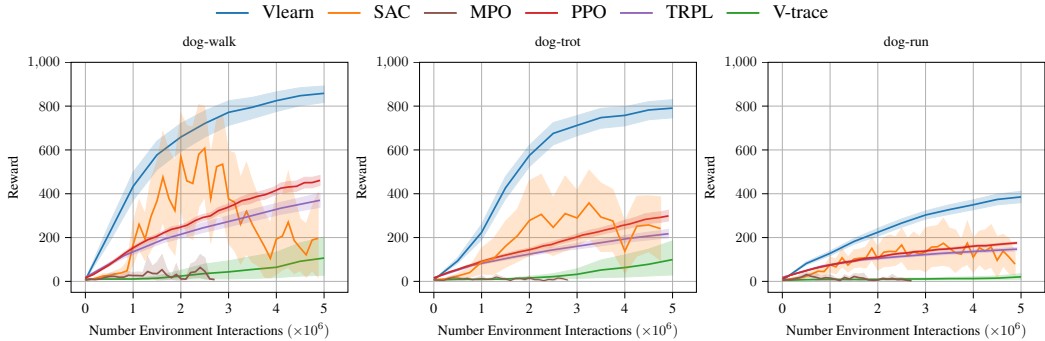

Figure 3: Performance on the 38-dimensional DMC dog tasks. Shown are the mean over 10 seeds and 95% bootstrapped confidence intervals. While SAC and MPO struggle to learn a consistent policy, Vlearn excels across all three tasks. On-policy methods show modest improvements and V-trace even struggles to make any meaningful progress.

it exhibits a slower convergence rate compared to SAC and seems to reach a lower local optima, similar to the on-policy methods. In general, HalfCheetah-v4 appears to be an outlier, challenging the learning capabilities of all trust-region methods. Even MPO, employing a related KL regularization concept, does not match the performance of SAC. Moreover, the environment itself has shown extreme behavior in the past (Zhang et al., 2021). However, for the other lower dimensional tasks Vlearn and MPO perform comparable to the other baseline methods.

In the case of the two higher-dimensional tasks, Ant-v4 and Humanoid-v4, Vlearn surpasses all other baselines in performance. Across both environments, Vlearn demonstrates better convergence speed and superior asymptotic performance. Particularly noteworthy is its remarkable performance boost for the Humanoid-v4, where Vlearn achieves a 25% increase over SAC. While in the case of Ant-v4, with its 8-dimensional action space, the improvement is more subtle, the advantage of Vlearn becomes evident in the significantly larger 17-dimensional action space of Humanoid-v4. In this scenario, focusing solely on learning a state-value function proves to be a less complex and more effective approach than attempting to learn the full state-action value function. Neither for the Ant-v4 nor for the Humanoid-v4 is MPO able to achieve a competitive performance.

In direct comparisons between Vlearn and its V-trace counterpart, Vlearn consistently outperforms V-trace across all tasks. Notably, the V-trace version hardly learns anything within the scope of a fixed number of environment interactions. Our investigations revealed that, while V-trace is eventually able to learn, it experiences significant drops in performance after an extended training period. Given that the sole difference between the two experiments lies in the V-trace objective, it is highly likely to be the source of this issue. The key distinction between our objective and V-trace lies in how they handle situations where the importance ratio approaches zero and the use of the expectations. In our objective, we assign these samples a weight close to zero, minimizing their influence on the gradient. Conversely, V-trace attempts to bring these samples closer to the target network, potentially leading to performance degradation. Additionally, estimating the joint expectation over states and actions is typically more stable. This observation aligns with findings in Mahmood et al. (2014); Dann et al. (2014), where it was demonstrated that importance sampling exclusively for the Bellman targets can lead to inferior performance. While they only evaluated this for the linear case, we find similar results for general non-linear function approximation.

## 4.2 DEEPMIND CONTROL SUITE

In addition to the Gymnasium tasks, we conducted experiments using the dog environments from the DMC suite. The dog tasks pose the most challenging problems in our evaluation, with a 38-dimensional action space modeling a realistic pharaoh dog. Equivalent to the Gymnasium setting, Vlearn exhibits superior performance on these high-dimensional dog locomotion tasks. Although we found that SAC benefits from layer normalization for these tasks, it struggles to learn well-performing policies for all three movement types. While SAC improves for "easier" movements, its convergence remains highly unstable, and the final performance often falls below that of on-policy

methods. MPO has similar problems as with the higher dimensional Gymnasium tasks and is unable to solve the tasks. In contrast, Vlearn reliably learns for all three distinct dog movement types.

Comparing our V-function learning approach to the V-trace estimator, the disparities are even more pronounced in for the DMC dog tasks. For these high-dimensional problems, the V-trace estimator fails to learn and is among the worst baselines, even falling behind on-policy methods. Generally, we observe a performance decline across the board, even in the case of lower-dimensional problems.

### 4.3 ABLATION STUDIES

In our ablation study, we investigate the effect of replay buffer size as well as the individual components of our method on the learning process. We trained multiple agents on the Gymnasium Humanoid-v4 with varying replay buffer sizes of $\{2e3, 1e4, 5e4, 1e5, 2.5e5, 5e5, 1e6\}$, for 10 different seeds each. It is worth noting, that while the original V-trace (Espeholt et al., 2018) relies on a relatively large replay buffer, the massively parallel computation used here implies the differences between the policy distributions is not as pronounced as in standard off-policy methods like SAC. As shown in Figure 2 (bottom right), significant improvement occurs when transitioning from updating the policy and critical setting near an on-policy setting, using the smallest replay buffer size, to using a medium buffer. Notably, transitioning from a small to a medium-sized buffer yields significant benefits, while little difference is observed between buffer sizes at the upper end of the investigated range. In our experiments, we found that a replay buffer size of $5e5$ consistently yielded optimal performance across all tasks and was also used in Figures 3 and 2.

For the second ablation study, we trained multiple variations of Vlearn for Humanoid-v4 to investigate the effects of the individual components (Figure 2). First, as naive baseline we removed the importance sampling (No IS) and assume all samples in the replay buffer are from the current policy. As expected not having the correction from importance sampling does not allow the agent to learn at all. When replacing the TRPL policy loss with the clipped PPO loss (PPO loss), our results suggest that the heuristic trust region provided by PPO is insufficient for the off-policy case where stabilizing learning is even more important. While PPO is able to achieve a decent performance on the task, its asymptotic performance lags behind the agent using TRPL. For the importance weight truncation, we followed previous works (Espeholt et al., 2018; Munos et al., 2016) by selecting $\epsilon_\rho = 1$ to reduce variance and potentially avoid exploding gradients. Similar to the PPO loss, we see that for a larger truncation level ($\rho_\epsilon = 20$) learning becomes unstable. Lastly, we trained an agent without the twin critic networks (No Twin), which performs significantly worse. We assume the twin network can be seen as a small ensemble that provides a form of regularization.

## 5 CONCLUSION AND LIMITATIONS

In this work, we have shown how to efficiently learn a V-function from off-policy data using an upper bound objective and use it to update the policy network. Our method of learning the V-function offers computational efficiency and enhanced stability and performance compared to existing approaches,. Combining this idea with the exact trust regions from TRPL yields an efficient off-policy method that performs particularly well for high-dimensional action spaces.

Although our method excels in handling high-dimensional action spaces, it may still require a substantial amount of data to achieve optimal performance. Hence, sample efficiency still remains a challenge that needs to be addressed further. Additionally, in specific environments, such as HalfCheetah-v4, Vlearn did not achieve a competitive performance compared to existing approaches. Improving the method's reliability, particularly in lower-dimensional scenarios, represents an ongoing challenge. For future work, we are looking to combine this work with other advances in RL, such as distributional critics (Bellemare et al., 2017) or ensembles (Chen et al., 2021). Furthermore, we are looking to extend our approach to the realm of offline RL, which offers the opportunity to leverage pre-collected data efficiently, opening doors to real-world applications and minimizing the need for extensive data collection.

## 6 REPRODUCIBILITY STATEMENT

In our experiments, we utilized standard RL benchmarks and followed the evaluation protocols from Agarwal et al. (2021). The pseudo-code of our method, implementation details, and hyperparameters used in our experiments can be found in Sections 3 and 4 and Appendix B. Readers can replicate our results by following the procedures outlined in these sections. Additionally, the complete source code for our reinforcement learning framework will be published alongside the camera-ready version of this paper.

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

## A    DERIVATION

We now show the proof of our new objective function. We start with a standard loss function for learning a V-value function for policy $\pi$ as defined in Equation 2. Then we derive its upper bound shown in Equation 4.

### A.1    UPPER BOUND OBJECTIVE

$$
\begin{aligned}
L^*(\theta) &= \mathbb{E}_{s \sim d^\pi(s)} \left[ \left( V_\theta^\pi(s) - \mathbb{E}_{a \sim \pi(\cdot|s)} \left( r(s,a) + \gamma V_{\bar{\theta}}(s') \right) \right)^2 \right] \\
&= \mathbb{E}_{s \sim d^\pi(s)} \left[ \left( V_\theta^\pi(s) - \mathbb{E}_{a \sim \pi_b(\cdot|s)} \left[ \frac{\pi(a|s)}{\pi_b(a|s)} \left( r(s,a) + \gamma V_{\bar{\theta}}(s') \right) \right] \right)^2 \right] \\
&= \mathbb{E}_{s \sim d^\pi(s)} \left[ \left( \mathbb{E}_{a \sim \pi_b(\cdot|s)} \left[ \frac{\pi(a|s)}{\pi_b(a|s)} V_\theta^\pi(s) \right] - \mathbb{E}_{a \sim \pi_b(\cdot|s)} \left[ \frac{\pi(a|s)}{\pi_b(a|s)} \left( r(s,a) + \gamma V_{\bar{\theta}}(s') \right) \right] \right)^2 \right] \\
&\quad (V_\theta^\pi \text{ does not depend on a}) \\
&= \mathbb{E}_{s \sim d^\pi(s)} \left[ \left( \mathbb{E}_{a \sim \pi_b(\cdot|s)} \left[ \frac{\pi(a|s)}{\pi_b(a|s)} \left( V_\theta^\pi(s) - \left( r(s,a) + \gamma V_{\bar{\theta}}(s') \right) \right) \right] \right)^2 \right] \\
&= \mathbb{E}_{s \sim d^\pi(s)} \left[ \left( \int \underbrace{\pi_b(a|s) \frac{\pi(a|s)}{\pi_b(a|s)}}_{\text{weight terms t}} \underbrace{\left( V_\theta^\pi(s) - \left( r(s,a) + \gamma V_{\bar{\theta}}(s') \right) \right)}_{x} da \right)^2 \right] \\
&= \mathbb{E}_{s \sim d^\pi(s)} \left[ f \left( \int tx \, da \right) \right],
\end{aligned}
$$

where we denote the convex function $f(x) = x^2$. While the weight terms are in $[0,1]$ and normalized, $\int t \, da = \int \pi_b(a|s) \frac{\pi(a|s)}{\pi_b(a|s)} da = \int \pi(a|s) da = 1$, the Jensen's inequality is applied as

$$
\begin{aligned}
f \left( \int tx \, da \right) &\leq \int t f(x) \, da \\
&= \mathbb{E}_{s \sim d^\pi(s)} \left[ \mathbb{E}_{a \sim \pi_b(\cdot|s)} \left[ \frac{\pi(a|s)}{\pi_b(a|s)} \left( V_\theta^\pi(s) - \left( r(s,a) + \gamma V_{\bar{\theta}}(s') \right) \right)^2 \right] \right] \quad \text{(Jensen's inequality)} \\
&= \mathbb{E}_{s \sim d^\pi(s), a \sim \pi_b(\cdot|s)} \left[ \frac{\pi(a|s)}{\pi_b(a|s)} \left( V_\theta^\pi(s) - \left( r(s,a) + \gamma V_{\bar{\theta}}(s') \right) \right)^2 \right] \\
&= L(\theta),
\end{aligned}
$$

where $d^\pi(s)$ is the stationary distribution induced by policy $\pi$.

Therefore, one can estimate $L(\theta)$ using Monte Carlo samples from the joint state-action distribution

$$
L(\theta) = \sum_t \sum_j \frac{\pi(a_{t,j}|s_t)}{\pi_b(a_{t,j}|s_t)} \left( V_\theta^\pi(s_t) - \left( r_{t,j} + \gamma V_{\bar{\theta}}(s_{t+1,j}) \right) \right)^2.
$$

### A.2    CONSISTENCY OF UPPER BOUND OBJECTIVE

We follow a similar derivation as in Neumann & Peters (2008) (for the case of state-action value function $Q(s,a)$) to prove the consistency between $L^*(\theta)$ and $L(\theta)$, i.e. the solution for minimizing

$L(\theta)$ is the same for the original objective $L^*(\theta)$. For simplicity, we denote $\bar{V} = r(s, a) + \gamma V_{\bar{\theta}}(s')$.

$$L^*(\theta) = \mathbb{E}_{s \sim d^\pi(s)} \left[ \left( V_\theta^\pi(s) - \mathbb{E}_{a \sim \pi(\cdot|s)}[\bar{V}] \right)^2 \right]$$

$$= \mathbb{E}_{s \sim d^\pi(s)} \left[ V_\theta^\pi(s)^2 - 2V_\theta^\pi(s)\mathbb{E}_{a \sim \pi(\cdot|s)}[\bar{V}] + \mathbb{E}_{a \sim \pi(\cdot|s)}[\bar{V}]^2 \right]$$

$$L(\theta) = \mathbb{E}_{s \sim d^\pi(s)} \left[ \mathbb{E}_{a \sim \pi_b(\cdot|s)} \left[ \frac{\pi(a|s)}{\pi_b(a|s)} \left( V_\theta^\pi(s) - \bar{V} \right)^2 \right] \right]$$

$$= \mathbb{E}_{s \sim d^\pi(s)} \left[ \mathbb{E}_{a \sim \pi_b(\cdot|s)} \left[ \frac{\pi(a|s)}{\pi_b(a|s)} \left( V_\theta^\pi(s)^2 - 2V_\theta^\pi(s)\bar{V} + \bar{V}^2 \right) \right] \right]$$

$$= \mathbb{E}_{s \sim d^\pi(s)} \left[ V_\theta^\pi(s)^2 - 2V_\theta^\pi(s)\mathbb{E}_{a \sim \pi_b(\cdot|s)} \left[ \frac{\pi(a|s)}{\pi_b(a|s)} \bar{V} \right] + \mathbb{E}_{a \sim \pi_b(\cdot|s)} \left[ \frac{\pi(a|s)}{\pi_b(a|s)} \bar{V}^2 \right] \right]$$

$(V_\theta^\pi \text{ does not depend on a})$

$$= \mathbb{E}_{s \sim d^\pi(s)} \left[ V_\theta^\pi(s)^2 - 2V_\theta^\pi(s)\mathbb{E}_{a \sim \pi(\cdot|s)} \left[ \bar{V} \right] + \mathbb{E}_{a \sim \pi(\cdot|s)} \left[ \bar{V}^2 \right] \right]$$

The above results show that $L^*(\theta)$ and $L(\theta)$ are identical except for a constant term added, which remains independent of $V_\theta^\pi$.

## B   HYPERPARAMETERS

Table 1: Hyperparameters for the Gymnasium (Towers et al., 2023) experiments in Figure 2. The larger sample size for the the two on-policy methods is for the Humanoid-v4 experiments.

| | PPO | TRPL | Vlearn/V-trace | SAC | MPO |
|---|---|---|---|---|---|
| number samples | 2048/16384 | 2048/16384 | 1 | 1 | 1 |
| GAE $\lambda$ | 0.95 | 0.95 | n.a. | n.a. | n.a. |
| discount factor | 0.99 | 0.99 | 0.99 | 0.99 | 0.99 |
| $\epsilon_\mu$ | n.a. | 0.05 | 0.1 | n.a. | 1e-3 |
| $\epsilon_\Sigma$ | n.a. | 0.0005 | 0.0005 | n.a. | 2e-6 |
| optimizer | adam | adam | adam | adam | adam |
| updates per epoch | 10 | 20 | 1000 | 1000 | 1000 |
| learning rate | 3e-4 | 5e-5 | 5e-4 | 3e-4 | 3e-4 |
| use critic | True | True | True | True | True |
| epochs critic | 10 | 10 | n.a. | n.a. | n.a. |
| learning rate critic (and alpha) | 3e-4 | 3e-4 | 5e-4 | 3e-4 | 3e-4 |
| learning rate dual | n.a. | n.a. | n.a. | n.a. | 1e-2 |
| number minibatches | 32 | 64 | n.a. | n.a. | n.a. |
| batch size | n.a. | n.a. | 64 | 256 | 256 |
| buffer size | n.a. | n.a. | 5e5 | 1e6 | 1e6 |
| learning starts | 0 | 0 | 0 | 10000 | 10000 |
| policy update interval | n.a. | n.a. | 2 | 1 | 1 |
| polyak_weight | n.a. | n.a. | 5e-3 | 5e-3 | 5e-3 |
| trust region loss weight | n.a. | 10 | 10 | n.a. | n.a. |
| num action samples | n.a. | n.a. | n.a. | 1 | 20 |
| normalized observations | True | True | True | False | False |
| normalized rewards | True | False | False | False | False |
| observation clip | 10.0 | n.a. | n.a. | n.a. | n.a. |
| reward clip | 10.0 | n.a. | n.a. | n.a. | n.a. |
| critic clip | 0.2 | n.a. | n.a. | n.a. | n.a. |
| importance ratio clip | 0.2 | n.a. | n.a. | n.a. | n.a. |
| hidden layers | [64, 64] | [64, 64] | [256, 256] | [256,256] | [256,256] |
| hidden layers critic | [64, 64] | [64, 64] | [256, 256] | [256,256] | [256,256] |
| hidden activation | tanh | tanh | relu | relu | relu |
| initial std | 1.0 | 1.0 | 1.0 | 1.0 | 1.0 |

Table 2: Hyperparameters for the dog tasks from DeepMind Control (Tunyasuvunakool et al., 2020) Figure 3.

|  | PPO | TRPL | Vlearn/V-trace | SAC | MPO |
|---|---|---|---|---|---|
| number samples | 16384 | 16384 | 1 | 1 | 1 |
| GAE $\lambda$ | 0.95 | 0.95 | n.a. | n.a. | n.a. |
| discount factor | 0.99 | 0.99 | 0.99 | 0.99 | 0.99 |
| $\epsilon_\mu$ | n.a. | 0.05 | 0.1 | n.a. | 1e-3 |
| $\epsilon_\Sigma$ | n.a. | 0.0005 | 0.0005 | n.a. | 1e-6 |
| optimizer | adam | adam | adam | adam | adam |
| updates per epoch | 10 | 20 | 1000 | 1000 | 1000 |
| learning rate | 3e-4 | 5e-5 | 1e-4 | 3e-4 | 3e-4 |
| use critic | True | True | True | True | True |
| epochs critic | 10 | 10 | n.a. | n.a. | n.a. |
| learning rate critic (and alpha) | 3e-4 | 3e-4 | 1e-4 | 3e-4 | 3e-4 |
| number minibatches | 32 | 64 | n.a. | n.a. | 1e-2 |
| batch size | n.a. | n.a. | 64 | 256 | n.a. |
| buffer size | n.a. | n.a. | 5e5 | 1e6 | 256 |
| learning starts | 0 | 0 | 0 | 10000 | 1e6 |
| policy update interval | n.a. | n.a. | 2 | 1 | 10000 |
| polyak_weight | n.a. | n.a. | 5e-3 | 5e-3 | 1 |
| trust region loss weight | n.a. | 10 | 10 | n.a. | 5e-3 |
| normalized observations | True | True | True | False | False |
| normalized rewards | True | False | False | False | False |
| observation clip | 10.0 | n.a. | n.a. | n.a. | n.a. |
| reward clip | 10.0 | n.a. | n.a. | n.a. | n.a. |
| critic clip | 0.2 | n.a. | n.a. | n.a. | n.a. |
| importance ratio clip | 0.2 | n.a. | n.a. | n.a. | n.a. |
| hidden layers | [64, 64] | [64, 64] | [512, 512] | [512,512] | [512,512] |
| hidden layers critic | [64, 64] | [64, 64] | [512, 512] | [512,512] | [512,512] |
| hidden activation | tanh | tanh | relu | relu | relu |
| initial std | 1.0 | 1.0 | 1.0 | 1.0 | 1.0 |

