# OpenReview forum: "Vlearn: Off-Policy Learning with Efficient State-Value Function Estimation"
_ICLR.cc/2024/Conference — Submitted to ICLR 2024_

### Official Review · Reviewer_WLtt · 2023-10-27

**Soundness:** 2 fair
**Presentation:** 2 fair
**Contribution:** 2 fair
**Rating:** 6
**Confidence:** 2

**Summary:**

Valued-based RL typically aims to learn the Q-function ("action value function"), which maps state-action pair to a Q-value. The complexity of this approach doesn't scale well with high-dimensional action space. Instead, this work proposes to learn the V-function ("state value function"). There's a some description of the mathematical foundation for the proposed loss function and algorithm.

Experiments were conducted on continuous control tasks including gym (mujoco) and DeepMind control suite.

**Strengths:**

- Strong empirical performance in the experiments, especially compared to V-trace, which shares a similar mathematical formulation as the proposed approach Vlearn.

**Weaknesses:**

- Lack of mathematical rigor in the algorithm description (see questions below)
- An important connection to a particular piece of prior work seems to be obscured in the paper as written

**Questions:**

- Issues with mathematical notation
  - In eqn (2) and the eqn right above eqn (4): what is $j$, what is the sum over $j$, and what is $K$? Below eqn (2), the text says "$a_{t,j}$ defines the $j$th action in state $s_t$". Does this mean $j$ is enumerating the collected transitions (presumably in a buffer)? Or is this following some other distribution - what distribution would that be?
  - In eqn (4), since you are no longer summing over $j$, do you still need $\frac{1}{K}$?
  - Right above eqn (3), could you clarify what "upper bound" and "lower bound" this sentence is referring to, and why "truncated importance weights can provide an upper bound"?
- The introduction mentions "upper bounds of actual Bellman error" - I assume this is referring to eqn (4) - however it was not explicitly stated around eqn (4).
  - How was Jensen's inequality used to derive eqn (4) from eqn (2)? Could walk through this explicitly.
- Connection to Mahmood et al. 2014
  - In that referenced paper, the authors considered linear function approx setting and presented two importance sampling approaches for off-policy learning.
  - What they call OIS-LS and WIS-LS seems to correspond to the difference between eqn (2) vs eqn (4). Specifically, their WIS-LS objective is $J(\theta) = \frac{1}{n} \sum_k \rho_k (Y - \theta^{\intercal} \phi_k)$, which basically corresponds to eqn (4) of this paper $L(\theta) = \frac{1}{K} \sum_{t} \rho_t (V_\theta(s_t) - (r_t + \gamma V_{\bar{\theta}}(s_{t+1}))^2$ except now $V_\theta$ is used in place of a linear function.
  - The core idea of the current proposed approach seems to be simply extending the linear case to general function approximation. Is there any other difference? Either way, this connection should be explicitly stated, perhaps citing the mathematical formulation as well. Of course, I understand the prior work did not consider any continuous control benchmarks, and the contribution of this work is still valuable. But I believe the connection should be made clear.
- The conclusion states "proposed a novel approach applying trust region methods in the off-policy setting" -- is that the main contribution? Or is learning V-function instead of Q-function the contribution.

Minor issues:
- typo on page 4 paragraph 2 "Bellmann" -> "Bellman"
- typo on page 6 "populate]"
- typo on page 8 conclusion "an novel approach" -> "a novel approach"

---

> ### Author Response · Authors · 2023-11-20
> **Rebuttal Answer for Reviewer WLtt**
>
> Dear Reviewer WLtt,
>
> Thank you for your thoughtful review of our paper, we appreciate the time and effort you dedicated to providing detailed feedback. We are pleased to note your recognition of the strong empirical performance demonstrated in our experiments, especially when compared to V-trace. Below, we address your open questions and concerns.
>
> - **Derivation Details and Theoretical Justification:** We acknowledge your concern about the ambiguity in the derivation of the upper bound for the value function. In the revised manuscript, we have improved the notation in the main text and provided a comprehensive derivation in the appendix. In addition, we have given more insight into the method, explaining its relationship to and differences from V-trace. We have also now included a full proof establishing the consistency of the upper bound.
> We hope that these revisions not only answer your questions about the derivation, but also provide a more intuitive understanding of the method. Should you require further clarification, we are available to provide additional information.
>
>   To answer some of the questions already at this point. The $\sum_j^K$ over actions in state $s_t$ is a Monte Carlo estimate of the expectation over actions from our policy. In practice computing this is hardly possible as we would have to be able to execute multiple actions from the same state. Our method now moves this expectation outside of the square and allows us to use the joint expectation over states and actions. But we agree this should be also reflected in the notation, we updated it accordingly.
>
> - **"Upper Bound" and "Lower Bound" Clarification Eq (3):** The upper bound here simply refers to the truncated importance weights that clip the importance weights at typically 1. While we do not leverage a specific lower bound, the importance ratio has a natural minimum at 0.
>
> - **Contribution and Connection to Mahmood et al. 2014:** Our work is indeed similar to this previous work and could be seen as an extension to the general case of function approximation. However, direct application of this solution is still challenging as seen in our added ablation study, hence we also propose a proper way on how to leverage these results in an efficient way as part of an off-policy trust region method. Although a direct application is not possible, we believe that this related work further supports our argument that using importance sampling for the full loss function is preferable. We have made this connection clearer in the revised version.
>
> - **Minor Issues:** Thank you for pointing out the typographical errors. In the revised manuscript, we corrected these inaccuracies.
>
> We believe that the newly added changes strengthen the overall clarity and robustness of our paper. Your feedback has been very helpful, and we remain at your disposal for any further inquiries.

---

> > ### Author Response · Authors · 2023-11-23
> > **Update about notation**
> >
> > Dear Reviewer WLtt,
> >
> > We wanted to inform you that we have now made significant changes to the derivation and notation in the paper and uploaded a second revision based on the feedback from the other reviewers. We hope this addresses your concerns as well.
> >
> > Thank you again for taking the time to review our paper.

---

### Official Review · Reviewer_aLJf · 2023-11-01

**Soundness:** 1 poor
**Presentation:** 2 fair
**Contribution:** 1 poor
**Rating:** 3
**Confidence:** 5

**Summary:**

This paper proposes a new objective to estimate the state value function in off-policy setting. The new objective modifies the importance-weighted value iteration objective by a) moving the importance weights to the outside of the squared loss and b) subsuming the per-action weights to the summation over the time horizon. This state value estimation method is then combined with many other RL practice, including using the target network, applying trust region constraint, truncating the importance ratios etc, to form a new algorithm. The paper empirically evaluates this new algorithm in control tasks.

**Strengths:**

The paper is well written, with a clear statement of the research question. The related work also summarizes well the on-policy and off-policy methods. The use of a trust-region method to stabilize the training and the policy update is also convincing and turns out be to working well in empirical experiments.

**Weaknesses:**

### Clarity & quality
**Important details on the value loss function are missing and unclear**:

The main contribution of this is the use of a surrogate objective for the value function learning. However, the deduction of this new surrogate objective lacks important explanations.
1. “accomplish this by relaxing the squared loss function from Equation 2 using Jensen’s inequality”. It is not straightforward to me how to use Jensen’s inequality to obtain $L(\theta)$. Why we can move the ratios just outside of the square?
2.  In Equation (4), the paper just subsumes the summation over $j$ into the sum over $t$. I’m not sure quite how this can be done, given that the summation over $j$ “implicitly assumes multiple action executions per state”. But the $L(\theta)$ in Equation (4) clearly does not have such implicit assumption. Further, in the subsequent sections, there is no discussion on this, and Algorithm 1 just presents a single-action estimate for the loss.

### Significance
**Some conclusions and statements may need to be further justified**
1. **the conclusion from the cited work may not carry over straightaway to the value estimation in this paper**. Specifically, “the new loss function is an upper bound of the original loss function and shares the same optima”. However, in the cited work (Neumann & Peters, 2008), the proof is for the value estimation of $\pi$ with a given Q and samples generated by the same policy (if I understand correctly). This is different from objective function in Equation (4), which is estimated using target networks and the off-policy samples.  Can the authors provide a theoretical proof for this convergence to the same optima? Also, regarding “solving the relaxed loss should yield results consistent with the original loss”, does it mean that the final loss will be consistent? Or the final policy will be the same?

2. **It is unclear how the proposed state-value-estimation can be useful for the algorithm**: Particularly, the proposed algorithm is a combination of the multiple independent ideas: state value estimation, target networks, trust region projection, Gaussian policy approximation, replay buffer and importance weight truncation. Any of these ideas can be crucial and contributive for the strong empirical performance. It would be hard to detach the state value estimation from the other components and claim that the proposed state value estimation is the key.

3. **Further empirical studies are needed to verify the proposed idea**: the main empirical results only show the evaluation reward/returns for the trained policies. They are insufficient to show if the proposed value estimation objective can be better in learning an accurate value estimate. Particular, those strong performance of Vlean can be (very likely) a result from other implementation tricks. The paper should consider using Monte Carlo estimate to compare the estimated values with Vlean and the groundtruth values. Moreover, the ablation studies on the replay buffer size are irrelevant to the main idea of the proposed value estimate and should be substituted with a more informative study, e.g., with/without the ratios when optimization $L(\theta)$. The current ablation studies give no clue on the state-value-function estimation.

**Questions:**

1. The paper states that “we aim to reduce the variance by replacing the standard importance sampling ratio with truncated importance sampling”. But Algorithm 1 has no such ratio clipping

Some language issues (those do not affect my assessment):
1. “This loss from Equation 5 can be optimized” Equation 5 is not a loss, but an objective to maximize.

---

> ### Author Response · Authors · 2023-11-20
> **Rebuttal Answer for Reviewer aLJf**
>
> Dear Reviewer aLJf,
>
> Thank you for your thorough review of our paper, we appreciate your time and the constructive feedback you provided. We are grateful for your positive feedback on the clarity of the research question, the well-written nature of the paper, and the comprehensive summary of related work. Additionally, we appreciate your acknowledgment of the convincing use of a trust-region method to stabilize training. We have carefully considered your questions and concerns and aim to address them in the revised version of our manuscript as well as in our answer.
>
> - **Derivation Details and Theoretical Justification:** We acknowledge your concern about the ambiguity in the derivation of the upper bound for the value function. In the revised manuscript, we have improved the notation in the main text and provided a comprehensive derivation in the appendix. In addition, we have given more insight into the method, explaining its relationship to and differences from V-trace. We have also now included a full proof establishing the consistency of the upper bound.
> We hope that these revisions not only answer your questions about the derivation, but also provide a more intuitive understanding of the method. Should you require further clarification, we are available to provide additional information.
>
>   To already address the question regarding the summation at this point, we indeed use a one sample estimate (as does V-trace), the formulation was probably misleading there. By moving the importance sampling out of the square in Vlearn, it allows us to work with the joint expectation over states and actions (this is what we referred to as subsummation). This way having only one action sample per state is less of an issue and leads to lower variance than having a one sample Monte Carlo estimate just for the expectation over actions, such as in V-trace. This is similar to what SAC is doing by only generating one action sample for its expectation over the Q-function. V-trace effectively splits the expectations of states and actions and still uses a one sample estimate for the expectation over actions in Eq. (2), leading to potentially higher variance.
>
> - **Utility of State-Value Estimation:** We agree with the reviewer that the effects of the individual components are currently difficult to filter out. We have extended the ablation section of the paper by performing additional variations of our proposed method. However, it should already be noted here that in our experiments we already compare to V-trace in an equivalent setting, using the same techniques and only changing the way the value function is trained. Nevertheless, V-trace is not able to achieve the same performance. For this reason, we also think that a comparison with a Monte Carlo estimator is difficult, since V-trace is not able to learn a reasonable policy, which would also be reflected in a relatively poor estimate of V-trace in this comparison.
>
> - **Language & Clarity:** We acknowledge there are discrepancies between Algorithm 1 and the method as stated in the main text. In the revised manuscript, we improved the algorithm description to make it more self-explanatory and aligned. We also fixed the mentioned language issues.
>
> We sincerely appreciate your thoughtful review and are committed to enhancing the clarity, soundness, and presentation of our work based on your suggestions. We look forward to your feedback on the revised version that aims to address the mentioned points.

---

> > ### Comment · Reviewer_aLJf · 2023-11-21
> > **Post-rebuttal feedback**
> >
> > Thank you for the response. I think the reponse has not answered my questions directly. I would expect the authors to give a bit more rigorous analysis on the upper bound. The new empirical results in the updated manuscript also did not meet my expectqtions. I thus keep my score unchanged.

---

> > > ### Author Response · Authors · 2023-11-22
> > > **Re: Post-rebuttal feedback**
> > >
> > > In response to your initial review, we have included a detailed derivation of the upper bound, a consistency proof based on Neumann & Peters, 2008, tailored to our specific case, and an extensive ablation study. This study highlights the impact and benefits of various elements of our final proposed method, for example, showing the performance differences between using the exact same algorithmic setup but with V-trace or without importance sampling.
> > >
> > > However, we recognize that our interpretation may not have fully met your expectations. Could you please provide further guidance on the specific details or aspects you were expecting?
> > >
> > > Thank you very much for your time and consideration.

---

> > > > ### Author Response · Authors · 2023-11-23
> > > > **Re: Re: Post-rebuttal feedback**
> > > >
> > > > After the feedback from Reviewer syZi, we now uploaded a new revision where we incorporate even more detailed steps for the application of Jensen's inequality. We trust that these enhancements contribute to a clearer understanding of how we arrived at our solution.

---

### Official Review · Reviewer_FMyp · 2023-11-07

**Soundness:** 3 good
**Presentation:** 2 fair
**Contribution:** 3 good
**Rating:** 6
**Confidence:** 3

**Summary:**

This paper provides an off-policy trust region method that using only V to learn the optimal policy. In details, this paper proposes some modification (an uppper bound learning objective) to V-trace, and learns the policy by TRPL. Vlearn is claimed to bring more efficient exploration and exploitation in complex tasks with high-dimensional action spaces. This paper tests Vlearn on both Gymnasium and DMC dog tasks, Vlearn achieves strong results across all tasks, especially on 38-dimensional dog tasks.

**Strengths:**

- Empirical results are strong, Vlearn achieves strong results on high-dimensional gym tasks and DMC dog tasks.
- Vlearn extends ppo (or TRPL) to off-policy methods, which is of good significance.

**Weaknesses:**

- Why does the learning objective in Vlearn work better than 1-step V-trace is still not clear to me. Despite the empirical results, it will be more sound if the author could give some (theoritical) analysis to it.
- Although Vlearn only needs to learn V so as to bypass the problem in explicitly learning Q function representation, it seems will bring the  problem to the estimate of $\frac{\pi(a|s)}{\pi_b(a|s)}$ if the action space is high-dimensional.
- Empirical results on easy, low-dimension task such as hopper, walker, cheetah are low.

**Questions:**

see weakness.

---

> ### Author Response · Authors · 2023-11-20
> **Rebuttal Answer for Reviewer FMyp**
>
> Dear Reviewer FMyp,
>
> Thank you for your detailed review of our paper, we appreciate your time and thoughtful evaluation of our work. We are particularly encouraged by your positive feedback on the strong empirical results and the significance of our work. Below we aim to address the remaining questions and concerns.
>
> - **Comparison to V-trace:** By moving the importance sampling out of the square in Vlearn, it allows us to compute the joint expectation over states and actions. This way having only one action sample per state should generally lead to less variance. V-trace effectively keeps the split expectations of states and actions and still uses a single sample to estimate the expectation over actions in Eq. (2), leading to potentially higher variance. Furthermore, V-trace is interpolating between the current (target) value function and the 1-step Bellman target
> $$
>     J(\theta) = E_{s_t \sim \mathcal{D}} \Bigg[ \Big( V_\theta^\pi(s_t) -
>     \Big(
>     (1-\rho_t) V_{\bar{\theta}}(s_t) + \rho_t \big(r_{t}+ \gamma V_{\bar{\theta}}(s_{t+1})
>     \big)
>     \Big)
>     \Big)^2
>     \Bigg].
> $$
> The smaller the importance ratio, the more the value function optimization focuses on the current (target) value function. However, for these samples, the overall loss scale and consequently the gradient remain unchanged. In the case of Vlearn, it is different – samples with small importance ratios contribute less to the total loss but still work towards the Bellman target. V-trace was initially designed for correcting off-policy asynchronous processing, where policy differences (i.e. importance ratios) are typically limited, so this is not much of a constraint. However, in a complete off-policy scenario, the samples in the replay buffer and the current policy deviate more rapidly and significantly from each other. We tried to emphasize this insight more in the revised version to make the distinction clearer.
>
> - **Estimation of Importance Sampling Ratio:** First of all, we want to mention that we can compute the importance ratio directly from our predicted Gaussian distributions in closed form, hence, unlike the Q function, it does not need to be estimated. While in higher dimensional action spaces the two Gaussians will naturally overlap less and approach 0 faster, we found that the state-wise trust regions from TRPL help stabilize this as the policy cannot move arbitrarily fast from any previous policy distribution. This ensures there are always sufficiently many samples in the replay buffer for which the ratio is $>0$.
> However, we are not 100% sure, we understand the reviewer's question regarding this correctly. If our provided answer is not sufficient, please feel free to follow up on this.
>
> - **Low-Dimensional Tasks:** We agree that our performance does not provide significant improvements for the low-dimensional task, but we did not expect it to, since the added complexity of learning a Q-function is likely negligible in these cases. Therefore, similar performance is a desirable result for our method. SAC on the HalfCheetah-v4, we see as an outlier, as other off-policy methods, such as MPO, also do not perform as well on this task, and it has been shown in previous work [1] that this particular task exhibits extreme behavior.
>
> Thank you once again for your review, and we look forward to your feedback of the revised version that incorporates your suggestions.
>
> ### References
> [1] Baohe Zhang, Raghu Rajan, Luis Pineda, Nathan Lambert, Andr ́e Biedenkapp, Kurtland Chua,
> Frank Hutter, and Roberto Calandra. On the importance of hyperparameter optimization for
> model-based reinforcement learning. In International Conference on Artificial Intelligence and
> Statistics, pp. 4015–4023. PMLR, 2021

---

> > ### Author Response · Authors · 2023-11-23
> > **Update about revision**
> >
> > Dear reviewer FMyp,
> >
> > We wanted to inform you that we have now made significant changes to the derivation and notation in the paper as well as provide more intuition about the inner workings of our method in comparison to V-trace.
> > In response to the feedback received from other reviewers, we have now uploaded a second revision. We believe that these revisions not only enhance the clarity of our presentation but also directly address the concerns you raised.
> >
> > Thank you again for taking the time to review our paper.

---

### Official Review · Reviewer_syZi · 2023-11-08

**Soundness:** 2 fair
**Presentation:** 3 good
**Contribution:** 2 fair
**Rating:** 5
**Confidence:** 3

**Summary:**

This paper proposes a novel off-policy actor-critic reinforcement learning method called Vlearn that only learns a state-value function for advantage estimation to mitigate the curse of dimensionality in tasks with large action spaces. The algorithm is based on a state-value function loss upper bound of one that multiplies the importance sampling ratio with the Bellman target. Vlearn leverages an experience replay memory for data efficiency and employs delayed update, double-Q-learning-like training, importance sampling ratio truncation, and trust region optimization for stability.

The experiments done on the Gymnasium and DMC environments demonstrate a competitive performance of Vlearn on most of the continuous control tasks. Vlearn exhibits an impressive performance on the DMC dog tasks that previous methods struggled to learn. The empirical study concludes with an ablation study on the effect of the size of the replay buffer on the algorithm.

**Strengths:**

- The motivation for this work is easy to understand. The authors straightforwardly introduce the problem they are trying to solve and explain why it is important.
- The background material is abundant and comprehensive, crediting the proper related works.
- The experiments demonstrate the strength of VLearn in high-dimensional action space tasks compared to a fair selection of competitors.

**Weaknesses:**

- The derivation skips directly to the final equation, not providing enough detail. It's also possibly erroneous. Please see the question section.
- Some equations and parts of the pseudocode are difficult to understand due to poor explanation of the notations. I will point out some examples in the question section.
- The ablation study only investigates the effect of the size of the replay buffer. However, Vlearn uses multiple techniques from previous works, e.g., importance sampling ratio truncation. Each of them may contribute to the reported final performance. A more comprehensive ablation study can definitely enhance the value of this work.

**Questions:**

- I tried to justify how the authors arrived at equation 4 from equation 2 using Jensen's inequality but failed. I hope the authors can explain how Jensen's inequality is applied here. The importance sampling ratios do not necessarily sum to one. Thus, the inequality may not hold.
- Some notations are used before definition. For example, I was unsure what $K$ represents until I got to the pseudocode to find out it was the mini-batch size.
- The pseudocode is somewhat vague. For instance, does the for $ i \in \\{1,2\\}$ indicate a for-loop or a random sample from the set in lines 9 and 13? What does $\theta$ in line 11 represent? It's not in the input to the algorithm.
- What is the motivation behind using two sets of parameters for the state-value function? The overestimation bias exists in Q-learning. I don't think it exists in the authors' formulation.

---

> ### Author Response · Authors · 2023-11-20
> **Rebuttal Answer for Reviewer syZi**
>
> Dear Reviewer syZi,
>
> Thank you for taking the time to review our paper. We are pleased to note your positive remarks on the clarity of our motivation, the comprehensive background, and the strength demonstrated in our experiments. Further, we appreciate your valuable feedback and constructive comments. Below, we address each point to ensure clarity and address any outstanding issues.
>
> - **Derivation Details and Theoretical Justification:** We acknowledge your concern about the ambiguity in the derivation of the upper bound for the value function. In the revised manuscript, we have improved the notation in the main text and provided a comprehensive derivation in the appendix. In addition, we have given more insight into the method, explaining its relationship to and differences from V-trace. We have also now included a full proof establishing the consistency of the upper bound.
> We hope that these revisions not only answer your questions about the derivation, but also provide a more intuitive understanding of the method. Should you require further clarification, we are available to provide additional information.
>
> - **Pseudocode:** Here, the "for" is just an indicator that we compute this loss for both critic networks and equivalently for the target critics in line 13, i.e. a for loop over critics. We decided on this notation since it is also commonly used in other works, e.g. SAC [1]. The $\theta$ in line 11 is indeed inconsistent, and should be the minimum of the two critic estimates. Overall, for the updated version, we have aligned the pseudocode with the main text.
>
> - **Overestimation Bias:** We agree the overestimation bias is not a main concern in our setting as compared to the Q-function based setting. We found, however, in practice that using two networks is also beneficial in our case (see new ablation study). We assume the second network can be seen as a small ensemble that acts as a form of regularization and by selecting the minimum value, we avoid introducing any overestimation bias by training two networks. As the twin network approach is common practice for the baseline methods, we kept it also in our case.
>
> - **Effect of Individual Components:** We agree with the reviewer that an ablation study on the effects of the individual components can be of good value. We extended this section of the paper by running additional experiments. However it should already be noted here that in our experiments we compare to V-trace in an equivalent setting using the same techniques and only changing the way the value function is trained. Yet, V-trace is not able to achieve the same performance.
>
> We sincerely appreciate your time and thoughtful consideration of our work. Should you have any additional questions or require further clarification, we are readily available to assist. Thank you once again for your valuable feedback and engagement.
>
>
> ### References
> [1] Tuomas Haarnoja, Aurick Zhou, Pieter Abbeel, and Sergey Levine. Soft actor-critic: Off-policy maximum entropy deep reinforcement learning with a stochastic actor. In International conference on machine learning, pp. 1861–1870. PMLR, 2018.

---

> ### Comment · Reviewer_syZi · 2023-11-22
> **Comment on the Revision and Follow-up Questions**
>
> Thank you for the update. The current version is certainly more readable than the previous version. Though not as comprehensive as I expected, the ablation study revealed some contribution of each component to the algorithm's performance.
>
> I have some follow-up questions and comments on this version of the work.
> - Regarding equation (2), the authors claimed ``Directly optimizing this can be difficult as approximating the inner expectation with Monte Carlo samples requires multiple action executions per state, an unrealistic assumption for most RL problems." I wonder why multiple action execution is needed to estimate the inner expectation. The state $s_t$ is sampled from $\mathcal{D}$. The action is also sampled from $\mathcal{D}$. Do they come from the same transition $(s_t, a_t, r_t, s_{t+1})$? In addition, the next state $s_{t+1}$ is sampled from $\mathcal{T}$. Does that mean we should sample $s_{t+1}$ by running the sampled $s_t, a_t$ in the environment? An analogous question goes to equation (4).
> - What is the difference between the loss in equation (4) and the loss for the off-policy semi-gradient TD(0) (Sutton and Barto, 2018, pp. 258), except for the use of a target network?
> - The derivation of the upper bound objective now includes more details but again skips Jensen's inequality part. It is not trivial to show how to move the importance sampling ratio out of the square. Let $\delta (s, a)$ denote the TD error, $\rho(s, a) = \pi(a|s)/b(a|s)$ denote the importance sampling ratio. The inner expectation of $L^*(\theta)$ is $E_b[(\rho(s,a)\delta(s,a))]^2$. If we directly apply Jensen's inequality, we get $E_b[(\rho(s,a)\delta(s,a))]^2 \le E_b[(\rho(s,a)\delta(s,a))^2]$.  $\rho(s,a)$ is trapped in the square. I assume the other reviewers are concerned with this step as well. Instead, I believe it's better to apply Jensen's inequality early on and add the importance sampling ratio afterwards: $E_\pi [\delta(s,a)]^2 \le E_\pi [\delta^2(s,a)] = E_b[\rho(s,a) \delta^2(s,a)]$.
>
> Given the current state of the work, I am going to raise my rating slightly to reflect it.
>
> ---
> ## Reference
>
> Sutton, R. S., Bach, F., &amp; Barto, A. G. (2018). *Reinforcement learning: An introduction*. MIT Press Ltd.

---

> > ### Author Response · Authors · 2023-11-23
> > **Re: Comment on the Revision and Follow-up Questions**
> >
> > Thank you for your feedback on our rebuttal, based on your comments, we have made several revisions to improve our work.
> >
> > - The term "require" may come across as overly absolute, our primary concern lies in emphasizing that relying on a single Monte Carlo sample to approximate the inner expectation, as seen in methods such as V-trace, poses challenges. This estimation is prone to low quality and high variance. Achieving a more reliable estimate necessitates multiple samples, implying either multiple action executions per state or occurrences of the same state in the replay buffer. We now also added this to the paper.
> > - We acknowledge the potential for confusion regarding the use of $\mathcal{T}$. This choice of notation was influenced by the convention in SAC. In practice, $s_{t+1}$ is part of the transition data stored in the replay buffer. To reflect this, we have modified the relevant equations to accurately capture this representation.
> > - Similar to [1,2], Sutton's version primarily addresses the linear case. Further, our newly added ablation study reveals that a seamless transition from the idea in the linear to the non-linear case is not directly viable and additional modifications are required.
> > - We agree that applying Jensen's inequality first and applying the importance ratio is another possible way to derive this bound. We decided to add the importance ratio first as this allows us to derive the upper bound directly from Eq. (2) and makes the connection between the original objective and the upper bound more straightforward. We now uploaded a new revision where we incorporate additional steps for the application of Jensen's inequality. We trust that these enhancements contribute to a clearer understanding of how we arrived at our solution.
> > - Lastly, as the reviewer mentioned the ablation study could be more comprehensive. We would like to know about the specific ablations you believe would be relevant to enhance the study.
> >
> > We hope that these revisions and answers address your concerns and contribute to a more robust and comprehensible presentation of our work. We look forward to any further comments or suggestions you may have.
> >
> > ### References
> > [1] A Rupam Mahmood, Hado P Van Hasselt, and Richard S Sutton. Weighted importance sampling for off-policy learning with linear function approximation. Advances in neural information processing systems, 27, 2014.
> > [2] Christoph Dann, Gerhard Neumann, Jan Peters, et al. Policy evaluation with temporal differences: A survey and comparison. Journal of Machine Learning Research, 15:809–883, 2014.

---

### Author Response · Authors · 2023-11-20
**Improvements in the revised version**

We would like to thank all of our reviewers for their time and valuable, constructive feedback. In response to their insightful comments, we have carefully revised our work and uploaded an updated version. Please see below for a comprehensive list of changes.

- Added the detailed derivation of the upper bound using Jensen's inequality
- Added details how Jensen's inequality is applied
- Added proof for the consistency of the upper bound
- Added more intuition for our proposed method
- Added additional ablation study of the effect of various components of the algorithm
- Language and Grammar improvements

---

### Meta-Review · Area_Chair_Yqo7 · 2023-12-05

**Metareview:**

This work extends Weighted Importance Sampling - Least Squares (WIS-LS) from linear setting to the general function approximation case and demonstrate its efficacy in for high dimensional action space environments.

Strengths: The empirical study clearly shows the efficacy of the proposed method for high dimensional action space environments. The ablation study in one of those tasks confirms the necessity for some components to certain extent.

Weakness: The fundamental idea of this work is not novel -- it's an extension of WIS-LS from the linear RL literature to the deep RL setting. Despite that it is always nice to see ideas from linear RL actually work in deep RL, this lack of novelty requires more theoretical analysis to justify the contribution of this work. In particular, it would be nice to have a theoretical analysis on why the proposed objective outperforms the VTrace objective. Moreover, the reason that some of the additional techniques such as twined critic works in the proposed method remains mysterious. It would be nice to investigate why those components work in scaling WIS-LS to deep RL.

**Justification For Why Not Higher Score:**

Lack of in-depth theoretical analysis to justify the empirical improvement. The fundamental idea is not novel.

**Justification For Why Not Lower Score:**

N/A

---

### Decision · Program_Chairs · 2024-01-16

Reject